# Alveolar Hyperoxia and Exacerbation of Lung Injury in Critically Ill SARS-CoV-2 Pneumonia

**DOI:** 10.3390/medsci11040070

**Published:** 2023-11-01

**Authors:** Ahilanandan Dushianthan, Luke Bracegirdle, Rebecca Cusack, Andrew F. Cumpstey, Anthony D. Postle, Michael P. W. Grocott

**Affiliations:** 1NIHR Biomedical Research Centre, University Hospital Southampton NHS Foundation Trust, Southampton SO16 6YD, UK; luke.bracegirdle@doctors.org.uk (L.B.); r.cusack@soton.ac.uk (R.C.); a.cumpstey@soton.ac.uk (A.F.C.); a.d.postle@soton.ac.uk (A.D.P.); mike.grocott@soton.ac.uk (M.P.W.G.); 2Clinical and Experimental Sciences, Faculty of Medicine, University of Southampton, Southampton SO16 6YD, UK

**Keywords:** oxygen therapy, COVID-19, intensive care, hyperoxia, lung injury

## Abstract

Acute hypoxic respiratory failure (AHRF) is a prominent feature of severe acute respiratory syndrome coronavirus 2 (SARS-CoV-2) critical illness. The severity of gas exchange impairment correlates with worse prognosis, and AHRF requiring mechanical ventilation is associated with substantial mortality. Persistent impaired gas exchange leading to hypoxemia often warrants the prolonged administration of a high fraction of inspired oxygen (FiO_2_). In SARS-CoV-2 AHRF, systemic vasculopathy with lung microthrombosis and microangiopathy further exacerbates poor gas exchange due to alveolar inflammation and oedema. Capillary congestion with microthrombosis is a common autopsy finding in the lungs of patients who die with coronavirus disease 2019 (COVID-19)-associated acute respiratory distress syndrome. The need for a high FiO_2_ to normalise arterial hypoxemia and tissue hypoxia can result in alveolar hyperoxia. This in turn can lead to local alveolar oxidative stress with associated inflammation, alveolar epithelial cell apoptosis, surfactant dysfunction, pulmonary vascular abnormalities, resorption atelectasis, and impairment of innate immunity predisposing to secondary bacterial infections. While oxygen is a life-saving treatment, alveolar hyperoxia may exacerbate pre-existing lung injury. In this review, we provide a summary of oxygen toxicity mechanisms, evaluating the consequences of alveolar hyperoxia in COVID-19 and propose established and potential exploratory treatment pathways to minimise alveolar hyperoxia.

## 1. COVID-19

Severe acute respiratory syndrome coronavirus 2 (SARS-CoV-2), a novel member of the enveloped ribonucleic acid (RNA) beta coronavirus family, is the infectious agent causing the coronavirus disease 2019 (COVID-19) pandemic that has resulted in significant health, social, and economic burdens worldwide [1]. Severe COVID-19 disease is associated with male gender predominance, older age, metabolic syndrome, and obesity [2]. Moreover, some patients develop a multi-system disease process involving major organs with acute myocardial injury, acute kidney injury, haematological abnormalities, and intracerebral complications with prothrombotic tendencies [3]. A minority of hospitalised patients with severe COVID-19 pneumonia develop acute hypoxaemic respiratory failure (AHRF) necessitating intensive care admission and the initiation of mechanical ventilation to support adequate arterial oxygenation [4]. As the management of these critically ill patients continues to evolve, the pathophysiology of severe COVID-19 lung injury remains an intriguing phenomenon [5]. Although COVID-19 is defined as a single disease entity with a single causative agent, diverse clinical phenotypes may warrant individualised treatment approaches. These phenotypes are likely to reflect the complex host–virus interaction associated with SARS-CoV-2 infection, particularly the degree of host immunothrombotic response during and after the viral illness [6]. There is often a lag of 7–12 days between infection and the development of AHRF with SARS-CoV-2 infection, and antiviral strategies at this stage appear to offer no survival advantage [7,8]. Some patients go on to develop sustained hypoxaemic respiratory failure with prolonged hospitalisations.

## 2. COVID-19, Oxygen Therapy, and Alveolar Hyperoxia

Oxygen therapy is the mainstay for treatment of SARS-CoV-2-induced hypoxemia with the goal of maintaining arterial blood oxygen content and avoiding tissue hypoxemia. Respiratory support in deteriorating patients can be augmented with high-flow nasal oxygen (HFNO), continuous positive airway pressure (CPAP), or non-invasive positive pressure ventilation (NIV), and in severe cases, intubation and mechanical ventilation may be required [9]. COVID-19 patients who are mechanically ventilated may be profoundly hypoxemic and require prolonged periods of a high fraction of inspired oxygen (FiO_2_) [10,11]. Pulmonary toxicity caused by alveolar hyperoxia is a well-established phenomenon in both healthy and injured lungs [12]. However, it is unclear if alveolar hyperoxia due to oxygen therapy accelerates the initial insult from SARS-CoV-2 infection and, consequently, may be a driving factor in SARS-CoV-2-induced acute lung injury [13]. Moreover, adjunctive therapies to minimise alveolar hyperoxia or the beneficial effects of anti-inflammatory agents in mitigating the effect of hyperoxic lung injury have not been evaluated so far in this context.

In the United Kingdom (UK), approximately 50% of COVID-19 patients who were admitted to an intensive care unit (ICU) had severe hypoxic respiratory failure with a partial pressure of oxygen in arterial blood to fraction of inspired oxygen (PaO_2_/FiO_2_) ratio of <13.3 kPa (<100 mmHg), and in an Italian ICU case series, 22% of patients required an FiO_2_ of 0.85 on admission [10,11]. Conventional intensive care therapy for patients with this degree of lung injury includes mechanical ventilation with supraphysiological FiO_2_ to maintain relative normoxaemia. However, there is a strong association among endotheliopathy, coagulopathy, and immunothrombosis within the pulmonary microvasculature in COVID-19 patients with acute respiratory failure [14]. Consequently, the sustained hypoxaemia endured by some patients is likely due to a combination of increased shunt fraction from oxygen diffusion impairment from alveolar capillary occlusion and alveolar epithelial damage as a result of direct viral infection. The delivery of a high FiO_2_ in this scenario may lead to alveolar hyperoxia with little improvement in gas exchange or oxygen delivery to the distant tissues. Direct supraphysiological oxygen exposure can lead to alveolar oxygen toxicity causing alveolar endothelial and epithelial cell damage, including alveolar type-II (AT-II) cells [12]. Moreover, the consequent decrease in the alveolar partial pressure of nitrogen leads to reabsorption atelectasis, further exacerbating systemic hypoxemia (Figure 1) [15].

The main feature of hyperoxia demonstrated in in vivo models and isolated cell cultures of non-COVID-related hyperoxia is cell death, which may involve apoptosis or necrosis [16,17]. However, differentiating the proportional causality of hyperoxic lung injury versus lung damage from the initial insult, in this case, SARS-CoV-2 viral pneumonia, is often difficult. Post-mortem studies of patients who died with severe COVID-19 pneumonia have shown diffuse alveolar damage with endothelial injury, capillary congestion, immune cell recruitment, and varying degrees of interstitial pneumonia and eventual fibrosis [18]. These features are common to both COVID-19 acute respiratory distress syndrome (ARDS) and hyperoxic acute lung injury [19]. There is also evidence of AT-II cellular disruption attributed to enhanced cellular exhaustion, apoptosis, and hypertrophy [12]. Although difficult to clinically differentiate oxygen toxicity from other causes such as the initial insult or barotrauma, these findings are probably due to a combination of direct cellular damage caused by SARS-CoV-2 infection acting on the angiotensin converting enzyme 2 (ACE2) receptors on the AT-II cell surface, iatrogenic alveolar hyperoxia, or a combination of both processes [20].

Hyperoxic acute lung injury may partly explain the progression of some patients from relatively compliant to a non-compliant lung typically seen with COVID-19 ARDS. A similar phenomenon is also commonly seen in preterm infants with neonatal bronchopulmonary dysplasia (BPD), known to be exacerbated by oxygen toxicity, which may be mitigated by the administration of corticosteroids to the prenatal mother and the delivery of an exogenous surfactant to the infant once born [21]. Despite the common use of oxygen to treat hospitalised COVID-19 patients, optimal targets for arterial oxygenation or oxygen saturation are not yet defined and remain a contentious issue. Current recommended oxygen delivery targets for COVID-19 are not based on randomised trial data and are, in part, derived from other disease cohorts which may not be transferable [22]. The balance between likely competing harms of alveolar hyperoxia and systematic hypoxaemia with organ hypoxia and dysfunction is important but difficult to quantify. Nevertheless, in a survey of intensive care physicians from the UK, the majority adopted a permissive hypoxaemic target, with an arterial oxygenation of 7.1–9.0 kPa (53–67 mmHg), similar to those for patients with ARDS due to causes other than COVID-19 [23]. The controversy persists with regard to the optimal systemic oxygenation targets for mechanically ventilated intensive care patients [24,25]. On-going clinical trials comparing different systemic oxygen targets (conservative vs. standard) in ventilated patients are currently underway and may provide some insight into the best approach in critically ill patients but still may not be applicable to COVID-19 patients with extreme hypoxemia [26,27,28]. However, a post hoc subgroup analysis of COVID-19 patients from the HOT-ICU study suggests that targeting a lower PaO_2_ of 8kPa may be beneficial in COVID-19 ICU patients [29].

The progressive changes in the COVID-19 lung may be in part due to alveolar cell oxygen toxicity as a result of alveolar hyperoxia. Consequently, the standard management paradigm for COVID-19 patients with ARDS and AHRF needs urgent revaluation. Rather than arterial oxygenation targets such as arterial oxygen saturation (SaO_2_) or partial pressure of arterial oxygen (PaO_2_), oxygen delivery to the tissues may prove to be more informative. Here, we explore these elements, including the mechanisms of hyperoxic alveolar injury and several established and theoretical interventions that may mitigate these harms.

## 3. Consequence of Alveolar Hyperoxia and Systemic Hyperoxaemia

The consequences of hyperoxic organ toxicity have been evaluated in many animal studies since the 19th century. These studies present the organ-specific deleterious effects of hyperoxia in different animal models and experimental conditions with variations in the FiO_2_, barometric pressure, and duration of exposure. Although there were variations in tolerance, pulmonary toxicity was consistently reported with characteristic pathological changes [30]. The studies investigating the effect of alveolar hyperoxia on human lungs are primarily conducted on healthy humans, with non-injured lungs. As a result, the exact dose and duration of oxygen exposure for human pulmonary toxicity and lethality are largely unknown, especially in injured lung conditions such as viral pneumonia.

### 3.1. Ubiquitous Use of Oxygen May Be a Problem—Pre COVID-19 Studies

The duration of tolerance and the lethal dose of oxygen have been evaluated by several small and large animal studies of hyperoxia over the past two centuries [30,31]. There were considerable variations in the tolerance of pulmonary toxicity at various oxygen tensions and survival ability between animal species; mostly, high dose and prolonged exposure of an FiO_2_ between 0.90 and 1.0 are associated with acute respiratory failure and death [30]. The toxicity appears to be proportionate to the FiO_2_ [31]. When challenged with an FiO_2_ of 0.85–1.0, animal models of lower primates (rhesus monkeys, baboons, sooty mangabeys, and squirrel monkeys) fared better and survived longer, averaging around 8 days (range 3–17 days), than other small animals (rats, mice, guinea pigs, and birds) [30,32]. Moreover, susceptibility and the magnitude of pulmonary oxygen toxicity are variable even between individuals from the same species, suggesting an individual genetic predisposition [33]. Animal models also suggest that there is age-related differences in response to hyperoxia, where neonatal lung is more resistant to hyperoxia-induced lung injury compared with adult lungs, implying that the progressive development of innate immunity may contribute to hyperoxia-induced lung injury [34]. Pathologically, hyperoxic lung injury is characterised by diffuse alveolar epithelial and endothelial damage with exudative pulmonary oedema and capillary leakage very similar to the characteristic features of ARDS [35,36].

Although human hyperoxic challenge studies for prolonged periods are no longer ethically feasible, historical work demonstrated normobaric exposure to an FiO_2_ of 1.0 for 14 h led to substernal distress and pleuritic chest pain [37]. In a later study, a longer duration of exposure (30–74 h) resulted in the development of cough and progressive dyspnoea with an associated decline in vital capacity and diffusing capacity for carbon monoxide (DLCO) [38]. Abnormalities in tracheal mucociliary movement are evident after 3 h of exposure to an FiO_2_ of 0.90–0.95 [39]. Moreover, an FiO_2_ of more than 0.95 for 17 h leads to significant alveolar capillary leak with increased mediators of fibroblast recruitment and proliferation in healthy adults [40]. These limited numbers or normobaric hyperoxic human studies combined with translation from large primate and small animal studies support the potential for pulmonary toxicity and lethality in a normal uninjured lung, which in general, is associated with hyperoxia of an FiO_2_ > 0.70 beyond an exposure duration of 24 h [30,32]. However, more importantly, infected injured lungs may respond differently to hyperoxic challenges than a normal lung. Indeed, hyperoxia in legionella pneumonia increased lethality with accelerated apoptosis in rodent models [41]. The implications of combined insults of viral pneumonia and hyperoxia in the development and progression of acute lung injury is largely unknown.

Several oxygen intervention studies of mechanically ventilated or intensive care and critically ill patients investigating various oxygen targets of hyperoxaemia have been published to date. A systematic review of oxygen therapy of >16,000 hospitalised adult patients with acute illness (IOTA) concluded that liberal peripheral oxygen saturation (SpO_2_) targets beyond 94–96% are associated with increased in-hospital morality [42]. Despite this evidence, recent randomised controlled trials of patients with AHRF (HOT-ICU) and ARDS (LOCO2) suggest no clinical benefits from conservative arterial oxygen (PaO_2_) targets between 50 and 70 mmHg [43,44]. Further, larger studies are currently underway nationally, UK-ROX, and internationally, MEGA-ROX, indicating that this is an important research question with ongoing controversy [26,27,28]. Although these trials aim to accept a degree of permissive hypoxemia, they do not address the negative impact of alveolar hyperoxia in patients with severe hypoxemic respiratory failure requiring high fractional inspired oxygen.

### 3.2. Oxygen Toxicity Mechanisms

The cellular pathways leading to hyperoxia-mediated lung injury are complex and beyond the scope of this review. Briefly, lungs are vulnerable to oxidative damage, which is exacerbated during inflammatory conditions. Hyperoxia disrupts the normal physiological homeostatic balance and increases oxidative stress by inducing highly reactive mitochondrial oxidative stress mediators such as superoxide (O_2_^−^), hydrogen peroxide (H_2_O_2_), hydroxyl radicals (OH^−^), and peroxynitrite anions (ONOO^−^). Unopposed reactive oxygen species (ROS) lead to compromised cellular function with a predisposition of oxidative damage to deoxyribonucleic acid (DNA) material, lipids, and proteins [45]. Viral infections are also associated with increased oxidative stress [46]. Furthermore, imbalances in antioxidant mechanisms including the enzymes superoxide dismutase (SOD), catalase, and glutathione peroxidase and small defence molecules such as glutathione, ascorbic acid, and vitamin E, may result in the presence of increased oxidative stress mediators leading to further direct mitochondrial and cellular damage [47,48]. A deficiency in native plasma antioxidants is a common feature in COVID-19 infection and COVID-19 critical illness [49,50,51]. In healthy physiological states, the balance of oxidant and antioxidant is highly regulated and alterations in this equilibrium can lead to a proinflammatory state with a subsequent influx of inflammatory cells, the activation of cytokine cascades, and increased vascular permeability [52]. Alveolar hyperoxia with reduced native antioxidants in severe SARS-CoV-2 infection is likely to contribute to an overwhelming redox imbalance (Figure 2).

Molecular pathways leading to hyperoxic acute lung injury involves the activation of a multitude of signal transduction pathways of cellular homeostasis. In healthy physiological states, there is a balanced regulation of cell growth and cell death by several regulatory mechanisms of apoptosis and necrosis. Exposure to hyperoxia and the subsequent generation of ROS by nicotinamide adenine dinucleotide phosphate oxidase (NOX) 1 phosphorylation leads to changes in protein kinases, transcription factors and cellular apoptotic/necrotic pathways. Mitogen-activated protein kinase (MAPK) signalling cascades involving extracellular signal-regulated kinases (ERK1/2), c-Jun N-terminal kinase (JNK), and p38 kinase are all implicated in hyperoxic acute lung injury [53]. The role of these stress-activated protein kinases in both the upregulation and downregulation of hyperoxia-induced cell death has been studied extensively in various animal models of cellular hyperoxia [19]. Moreover, the hyperoxic exposure of lung stimulates molecules involved in the regulation of cell death, Fas and the Fas ligand with downstream activation of Caspase-8, and pro-apoptotic proteins (Bax, Bid, Bim, and Bak). A subsequent increase in protein kinase C delta type (PKC-δ) leads to the release of mitochondrial cytochrome C, and further cleavage of caspase-3 and 9 results in apoptotic and necrotic cell death [54]. Other transcription factors such as nuclear factor kappa B (NFκB), activator protein-1 (AP-1), signal transducer and activator of transcription (STAT), nuclear factor-erythroid 2-related factor 2 (Nrf2) and Toll-like receptor 4 (TLR4) play intricate counterbalanced roles in hyperoxic acute lung injury. The protective role of Nrf2 in antioxidant defence during hyperoxia is well established, and genetic polymorphisms in the Nrf2 gene may increase the epigenetic susceptibility to developing hyperoxic acute lung injury [55,56].

### 3.3. Alveolar Hyperoxia Induced Surfactant Damage

Pulmonary surfactant is a mixture of lipids and proteins at the air–liquid interface that minimises surface tension forces and prevents alveolar collapse. A surfactant is synthesised and metabolised by AT-II cells. Surfactant composition varies between animal species, but in humans, 70–80% of phospholipids are phosphatidylcholine (PC), with dipalmitoyl phosphatidylcholine (DPPC) being the major PC with the functional ability to reduce surface tension [57]. Surfactant deficiency is a commonly recognised feature in patients with ARDS, with variations in synthesis, metabolism, and functional inhibition from alveolar inflammatory milieu [58]. In vitro studies suggests that the surface activity of surfactant is impaired when directly exposed to oxidation [59]. A pulmonary surfactant exposed to oxygen free radicals can lead to the direct inactivation of phospholipids and proteins [60]. Moreover, in vivo animal studies showed that prolonged exposure to hyperoxia results in increased levels of oxidative stress with associated alterations in surfactant metabolism and turnover with reductions in lung antioxidant levels [61,62,63].

An intact surfactant system is fundamentally required for alveolar stability. SARS-CoV-2 pneumonia presents a double-edged sword, causing impairment to the surfactant system by AT-II cell death in combination with oxidative damage from alveolar hyperoxia [64]. Changes in AT-II cell proliferation and apoptosis are a common pathological feature of SARS-CoV-2 pneumonia. Alveolar epithelial cellular invasion by SARS-CoV-2 via ACE2 surface receptors causes increased AT-II cell apoptosis and can theoretically impair surfactant synthesis, secretion, metabolism, and recycling of functional pulmonary surfactant. However, the impact of SARS-CoV-2 infection on the surfactant system has yet to be explored. Alveolar hyperoxia may also compromise surfactant metabolism and the functional ability of surfactant to reduce surface tension resulting in poor lung compliance and worsening hypoxemia [65,66,67]. A reduction in overall surfactant pool size and functional ability to reduce surface tension is compromised during ARDS, and the same principle may apply during COVID-19 pneumonia [68].

Exogenous surfactant therapy has proved ineffective in adult patients with ARDS. A meta-analysis of exogenous surfactant replacement in ARDS patients demonstrated no survival benefit but some improvement in oxygenation within the first 24 h, which was not sustained [69]. This implies that the beneficial effect of a surfactant is possibly short-lived and a longer duration of therapy until recovery may need to be considered [70]. Moreover, a supplemented exogenous surfactant will likely face the same fate as an endogenous surfactant, with increased oxidative damage leading to a poor functional ability to maintain a low surface tension. An in vitro study of bovine surfactant suggested that surfactant performance is severely compromised when exposed to high concentrations of oxygen [68]. However, the effect of high FiO_2_ on surfactant composition, metabolism and functional surface tension-reducing ability in vivo, particularly in human injured models, has not been studied. Animal models of surfactant supplementation following hyperoxic exposure show an attenuation of alveolar oxygen toxicity, lung injury, and alveolar capillary permeability [71,72]. Surfactant also has antioxidant properties, and administration reduces the oxidative stress in animal models of hyperoxia [73,74,75]. From these limited available studies, it is tempting to speculate that an exogenous surfactant may be used as an antioxidant to minimise oxygen toxicity or in combination with other antioxidants to improve the surface-active properties of both endogenous and supplemented exogenous surfactants.

### 3.4. Alveolar Hyperoxia and the Expression of ACE2 Receptors

It has been postulated that the low incidence of COVID-19 among high-altitude inhabitants is possibly due to the downregulation of ACE2 expression during chronic hypoxia leading to fewer available receptors for viral entry [76]. However, the interaction between ACE2 expression and the risk of COVID-19 infection, progression into critical illness, and recovery is likely to be complex. Moreover, there are contradicting hypotheses as both hypoxia and hyperoxia seem to modify ACE2 expression in alveolar epithelia [77]. As detailed in an earlier section, the maintenance of normal ACE2 expression is crucially important for host immune response, and elevated levels of ANGII are associated with increased SARS-CoV-2 viral load and the severity of lung injury [78]. While increased ACE2 expression may increase the risk of viral infection, reduced ACE2 receptor availability can result in increased levels of ANGII and pulmonary fibrosis [79,80]. Infection with SARS-CoV-2 depletes host ACE2 and may contribute to the detrimental effects seen in the respiratory system. ACE2 protects against fibrosis, and in ACE2 knockout animal models, there is eventual development of severe lung disease and fibrosis [81,82]. An adult mice model of hyperoxia was associated with decreased lung ACE2 expression and increased ANGII/ANG (1-7) ratio, and co-treatment with ACE2 agonists mitigated the oxidative stress [83]. Likewise, hyperoxia downregulates ACE2 in human foetal fibroblasts, which may explain the development of BPD in neonatal lung disease [84]. However, the degree and duration of hyperoxia required to induce in vivo ACE2 modifications in humans is not known. All these findings suggest that a combination of SARS-CoV-2 infection and hyperoxia are likely to contribute to an overwhelming downregulation of ACE2 expression and increasing ANGII levels, leading to a potentially lethal lung disease with eventual fibrosis. However, the multiplatform REMAP-CAP trial in critically ill patients recently concluded that neither ACE inhibition nor direct angiotensin receptor block improve clinical outcomes [85].

### 3.5. Alveolar Hyperoxia Induced Pulmonary Vascular Changes

ACE2 receptors are also expressed by pulmonary vascular endothelial cells. Alveolar capillary endothelial invasion and subsequent vascular damage with changes in cellular morphology and apoptosis are suggestive of endothelial inflammation as a prominent aetiology for pulmonary organ dysfunction precipitating respiratory failure and death [14,86]. The morphological changes in hyperoxic acute lung injury are also associated with pulmonary vascular changes, with damage to the pulmonary capillary bed and capillary leak resulting in pulmonary oedema. Although alveolar hypoxia increases pulmonary vasoconstriction and pulmonary vascular resistance (PVR), and conversely, hyperoxia leads to pulmonary vasodilation and a reduction in PVR, prolonged periods of hyperoxia may lead to a blunted vasodilatory response to nitric oxide (NO). Moreover, prolonged hyperoxia may also cause oxidative stress, which impairs the vasodilatory effects of endogenous and inhaled nitric oxide (iNO), potentially altering the potential for vascular reactivity.

### 3.6. Hyperoxia Induced Immune Dysfunction and Secondary Bacterial Infections

In critically ill patients, hyperoxaemia is independently associated with ventilator-associated pneumonia [87]. Secondary bacterial and ventilator-associated pneumonia is relatively common in patients who are ventilated for COVID-19 pneumonia [88]. The commonly identified organisms are klebsiella pneumonia, pseudomonas, and staph aureus species [89]. Animal models of in vitro cultured macrophages when exposed to hyperoxia demonstrate compromised macrophage-driven innate immune functions [90,91]. Furthermore, hyperoxaemia is associated with an increased susceptibility to Gram-negative bacterial infection with increased mortality [92]. Moreover, the lung and gut microbiomes are greatly alerted in animal models of alveolar hyperoxia and in critically ill patients receiving high concentrations of oxygen therapy, predisposing to secondary bacterial infections [93,94]. The hyperoxia-induced impairment in the innate immunity may in part explain the increased incidence of nosocomial infections seen in mechanically ventilated COVID-19 patients.

## 4. SARS-CoV-2 Pathogenesis Suggest Significant Endotheliopathy, Coagulopathy, and Microangiopathy

ACE2 is expressed in all human organs and is the primary target for SARS-CoV-2 viral entry. In physiological states, ACE2 plays a crucial role in the homeostasis of the renin–angiotensin–aldosterone system and is involved in the conversion of angiotensin II (ANGII) to angiotensin (1-7) (ANG (1-7)). The opposing functional effects of ANGII and ANG (1-7) are normally tightly balanced and the accumulation of ANGII can cause intense vasoconstriction and reduced perfusion, leading to profound ventilation/perfusion mismatch and resultant hypoxia [95]. Moreover, in addition to its vasoactive properties, ANGII can induce plasminogen activator inhibitor 1 and 2, leading to prothrombotic tendencies, and the endovascular thrombosis is not limited to large vessels [96]. Indeed, in experimental animal models, ANGII induced microvascular thrombosis relating to abnormalities in the coagulation cascade involving both thrombin and platelet aggregation [97].

ANGII is also a potent stimulator of nicotinamide adenine dinucleotide phosphate (NADPH) oxidase, activating the generation of reactive oxygen and nitrogen species [98]. The opposing effects of ANG (1-7) acting on the mitochondrial assembly (Mas) receptor has several protective functions, including vasodilatation, as well as anti-inflammatory, antioxidant, and anti-thrombotic properties [99]. Binding of the SARS-CoV-2 spike protein S1 subunit to ACE2 receptors leads to cleavage of the receptor itself by a disintegrin and metallopeptidase 17 (ADAM17), also known as tumour necrosis factor alpha converting enzyme (TACE), at the cell surface, followed by intracellular cleavage by transmembrane protease, serine 2 (TMPRSS2) [100]. The eventual shedding and downregulation of ACE2 receptors leads to decreased endogenous ACE2 activity, which is pro-inflammatory with a state of increased oxidative stress (Figure 3 and Table 1) [101]. Alveolar compliance may be relatively preserved during this vascular phase with minimal contributions from alveolar flooding and epithelial or basement membrane disruption. Clinically, oxygen therapy is administered to mitigate hypoxemia during this stage, with consequent alveolar hyperoxia. Dual energy computed tomography (CT) studies of patients with early COVID-19 pneumonia have demonstrated significant perfusion abnormalities, often with relatively preserved lung parenchyma [102].

Moreover, the aberrant activation of the coagulation pathways supports the hypothesis of a vascular/endothelial process for severe COVID-19 pathogenesis [103]. Raised D-dimer levels are independently associated with increased mortality and hospital complications, and thrombocytopenia, which is often mild if present, is also associated with disease severity and poor clinical outcomes [104,105,106]. Markers of endothelial injury such as von Willebrand’s factor (VWF) and its cleaving protein, a disintegrin and metalloproteinase with thrombospondin type 1 motif, member 13 (ADAMTS13), also known as von Willebrand factor-cleaving protease (VWFCP), are also implicated in COVID-19 disease severity [107]. ADAMTS13 deficiency coupled with an increment in VWF multimers suggests profound endothelial dysfunction with acquired thrombotic microangiopathy [108]. Likewise, autopsy studies of patients having died from severe COVID-19 respiratory failure demonstrate an occlusion of alveolar capillaries with microangiopathy, microthrombi, and abnormal endothelial morphology, incriminating a strong link between SARS-CoV-2 infection, coagulation, and endothelial dysfunction [14,109,110]. Such complex coagulopathic processes need rapid clinical evaluation as standard anticoagulation or antiplatelet treatments are likely to be ineffective [111,112]. The evaluation of novel therapies such as recombinant ADAMTS13 replacement or nanobody molecules that inhibit the binding of VWF to platelet glycoprotein-1b receptors may be of value [113,114].

## 5. How to Minimise Alveolar Hyperoxia?

The dose and duration of oxygen therapy appears to be crucially important for the development of hyperoxic acute lung injury. This is particularly relevant in COVID-19 due to the prolonged periods of severe hypoxia in patients with COVID and current practices of supporting patients with severe hypoxia. Healthy human studies indicate a high FiO_2_ at 101.3 kPa (1 standard atmosphere (atm)) could be harmful [32,115,116]. Individual responses to hyperoxia vary, as demonstrated in animal models, which suggests a genetic predisposition of hyperoxic tolerance. Our proposal provides three potential solutions to minimise acute hyperoxic lung injury. First, ventilation and rescue strategies should be optimised to minimise prolonged hyperoxic exposure. Second, a more focused approach should be used to improve alveolar microangiopathy and immunothrombosis to facilitate adequate gas exchange. Third, although still an investigational approach, antioxidants should be evaluated as part of clinical trials to mitigate the effect of hyperoxia on the alveolar environment in patients requiring a high FiO_2_.

### 5.1. Permissive Hypoxaemia

Observational studies of critically ill patients indicate that the approach to oxygen therapy has been tolerating systemic hyperoxia in order to avoid hypoxemia [117,118]. Guidelines for the prescription of oxygen have recognised this, and now support targeting ‘normal’ or ‘near-normal’ oxygen saturations [22]. While specific oxygen delivery targets may be recommended in the resuscitation of acutely injured patients, hyperoxia does not seem to improve outcomes in critical illness and may even cause harm [24,119,120]. Targeting normoxia in critically ill patients without exposure to alveolar hyperoxia may not be practically achievable [121]. Oxygen therapy’s benefit-to-harm ratio is likely determined by the dose and duration, and so, strategies to reduce both are currently being investigated. Permissive hypoxaemia is a novel lung-protective strategy where lower that normal levels of arterial oxygenation are deliberately targeted, or tolerated, in an effort to minimise alveolar hyperoxia [122]. This is justified by the fact that humans have a variety of adaptive mechanisms that support hypoxia tolerance, perhaps demonstrated best at altitude [121,123]. However, more recent clinical trials of critically ill patients failed to confer any survival benefits with lower conservative oxygen targets [43,44,124]. Further larger studies are underway to establish safe lower limit for peripheral oxygenation targets [26,27,28].

Permissive hypoxaemia may also have some negative effects. Both cellular and organ adaptations to hypoxia play important roles in facilitating survival during critical illness, and intentionally tolerating hypoxia could result in further injury [125]. Hypoxia may also worsen or indeed cause pulmonary hypertension [126]. Another consideration is the avoidance of tissue ischaemia by improving oxygen delivery (DO_2_) and consumption (VO_2_). Goal-directed therapy through the dynamic assessment of oxygen indices and the optimisation of haemoglobin may be helpful when implementing permissive hypoxemia [122]. In patients who have had sufficient time to adapt to sustained hypoxaemia, permissive hypoxaemia may be a viable approach to reducing alveolar and systemic hyperoxia [121]. The harm of hypoxia should be weighed against the reduction in hyperoxic injury on an individual patient basis. As no specific threshold defines a safe level of permissive hypoxaemia, this remains an important unanswered research priority [24,25,127]. Nevertheless, permissive hypoxaemia was a common practice adopted widely in the UK during the COVID-19 pandemic and a post hoc analysis of a HOT-ICU study suggested better outcome with lower systemic oxygen targets in COVID-19 patients [23,29].

### 5.2. Adjunctive Measures to Optimise Oxygenation

#### 5.2.1. Mechanical Ventilation 

Although life-threatening refractory hypoxaemia is not frequently encountered in the generic ICU population, the COVID-19 pandemic resulted in frequent exposure to such patients and ventilator care bundles should incorporate strategies to minimise acute hyperoxic lung injury. The additional aim of a ventilator care bundle should incorporate measures to minimise the FiO_2_ to less than 0.70, but with an overall objective of achieving the lowest concentration with the minimum duration where possible. The open lung strategy with low tidal volume, high positive end-expiratory pressure (PEEP), with permissive hypercapnia targeting a plateau pressure of ≤30 cmH_2_O and driving pressure of ≤15 cmH_2_O is often used in ARDS, which may be transferable to COVID-19 patients [128]. Non-conventional ventilation modes such as airway pressure release ventilation (APRV), which is a time-triggered, pressure-targeted mode of ventilation, are associated with a reduced duration of mechanical ventilation, ICU length of stay, hospital mortality, and improvement in oxygenation in patients with ARDS [129]. Compared with standard ARDS management, there was an increased use of APRV during the COVID-19 pandemic in the UK [23]. Further trials are needed to draw conclusions of the benefits of APRV in COVID-19-related AHRF.

#### 5.2.2. Pulmonary Vasodilators

The use of pulmonary vasodilators in the forms of iNO or prostaglandin analogues may help to reduce shunting due to hypoxic vasoconstriction and may improve ventilation-perfusion matching. Despite ongoing controversy, pulmonary vasodilators are often used as part of the rescue strategy in patients with ARDS and COVID-19 AHRF [11,23,130,131,132,133,134,135]. In a survey of intensive care physicians across the UK, iNO and prostaglandin analogues as pulmonary vasodilators were used for treating COVID-19 patients by around 20% and 45% of responders, respectively [23]. However, the use of inhaled pulmonary vasodilators can be limited due to the lack of availability of delivery devices. Inhaled prostaglandins are straightforward to deliver and have been shown to improve oxygenation in patients with ARDS [132]. Moreover, inhaled pulmonary vasodilators may have additional antiviral, anti-inflammatory, and antiplatelet aggregation properties, with a potential to minimise disease burden [136,137,138]. Although guidelines do not support the routine use of inhaled pulmonary vasodilators in ARDS, multiple observational studies are published in this area, suggesting improvement in oxygenation, and multiple randomised controlled trials are planned or ongoing in COVID-19 patients.

#### 5.2.3. Prone Positioning

Prone positioning in ARDS patients has strong evidence for improving mortality [139]. A systematic review of observational cohort studies of both spontaneously breathing and mechanically ventilated COVID-19 patients demonstrated a significant improvement in oxygenation following prone positioning [140]. Although there are no randomised controlled trials to date in ventilated COVID-19 patients, the Proning Severe ARDS patients (PROCEVA) multicentre trial demonstrated a significant mortality benefit with 16 h of proning in ARDS patients with an FiO_2_ > 0.6 and a PaO_2_/FiO_2_ ratio of 20 kPa (<150 mmHg) [141]. Until further evidence emerges from randomised controlled trials, the same principles should apply for all mechanically ventilated COVID-19 patients with AHRF. Attempts should be made to self-prone as much as possible in all awake spontaneously breathing patients, and a minimum of 16 h of proning should be considered in all mechanically ventilated patients requiring a high FiO_2_ in efforts to minimise hyperoxic lung injury [141,142].

#### 5.2.4. Extracorporeal Membrane Oxygenation

The role of extracorporeal membrane oxygenation (ECMO) in refractory hypoxaemia due to ARDS and influenza A virus subtype H1N1-related ARDS is well established [143,144]. In the UK, ECMO services are commissioned by the National Health Service (NHS) England and provided by only a few established tertiary centres. It is a useful tool and often regarded as a last resort for refractory hypoxaemia in patients with reversible lung disease. The early use of ECMO has the potential to minimise not only barotrauma but also hyperoxia-induced acute lung injury. There are specific guidelines and referral pathways which are aimed to facilitate early referrals to avoid irreversible lung damage. The survival rates of ECMO-treated COVID-19 patients are like that of non-COVID ARDS, and where available, early referral is recommended for those with potentially reversible severe respiratory failure following an unsuccessful trial of lung-protective ventilation and prone positioning [145]. Equally, the provision of ECMO across individual centres will depend on the accessibility of this specialist service and the ability to expand the service capacity during pandemic situations.

### 5.3. Improving the Burden of Microangiopathy and Microthrombosis—Potential Targets

Current evidence does not support the use of augmented or therapeutic anticoagulation in the context of severe COVID-19 critical illness. Although the microvascular changes are likely due to a combination of in situ thrombosis and thromboembolic phenomena, data from randomised controlled trials suggest that the intermediate dosing of anticoagulation (e.g., enoxaparin 1 mg/kg once daily) in the intensive care unit setting and therapeutic anticoagulation with rivaroxaban or enoxaparin in non-critically unwell hospitalised patients in comparison with standard thromboprophylaxis does not improve clinical outcomes [146,147]. From the existing clinical trials, the National Institute for Health and Care Excellence (NICE) UK provided guidance and a conditional recommendation for the use of therapeutic anticoagulation in young, hospitalised patients on low-flow oxygen and without an increased bleeding risk [148]. Several antiplatelet therapies including aspirin and clopidogrel are currently being evaluated in clinical trials. However, the coagulation cascade targets are likely to be more complex [112]. The ADAMTS13-VWF-platelet axis needs further exploration and is not modified by standard anticoagulation or antiplatelet therapy. The augmentation of circulating ADAMTS13 by therapeutic supplementation or the reduction of large VWF multimers may reduce the alveolar microangiopathic process, which has not yet been explored [113]. Moreover, the recent use of nanobody technology to treat thrombotic thrombocytopenic purpura (TTP) may be of value, which binds to the A1 domain of the VWF, blocking its interaction with the platelet glycoprotein-Ib-IX-V receptor and therefore preventing platelet aggregation [149].

### 5.4. Improving Oxidative Stress and Inflammation—Exploratory Targets

In some critically ill patients, despite all appropriate ventilation and rescue strategies, the use of oxygen beyond an FiO_2_ of >0.70 is inevitable. So far, there are no effective pharmacotherapies available that moderates acute severe hyperoxic lung injury in humans. Although epigenetic susceptibility to acute hyperoxic lung injury cannot be modified, conceptually, the use of antioxidants in this scenario could potentially mitigate predictable oxidative lung damage. Potential therapeutic targets may modify the molecular pathways of hyperoxia-induced lung injury or augment antioxidant balance/capacity. However, currently, there are no specific treatments available that moderate or prevent hyperoxic lung injury. Non-COVID 19 mice models of vitamin E deficiency have been shown to exacerbate hyperoxic lung injury, and both vitamin E and C have protective effects on hyperoxia-induced cell death in human airway epithelial cells [150,151]. Moreover, vitamin E and C may have a protective role against the hyperoxia-induced denudation of cilia from airway epithelial cells [152]. N-acetyl cysteine is a precursor of glutathione and an antioxidant that has been used both in nebulised and intravenous forms to mitigate hyperoxic cytopathic effects in animal models [153,154]. Pulmonary surfactant components have antioxidant activity, and exogenous surfactant supplementation can attenuate hyperoxic lung injury in adult mice [72,73,155]. Transgenic mice models also suggest that the over expression of surfactant protein D (SP-D) has a protective role against hyperoxic lung injury via the upregulation of Nrf2 expression [156]. Although this list is not exhaustive, additional measures to augment global alveolar antioxidant status in those with intractable hypoxaemia requiring an FiO_2_ of >0.70 may improve the local alveolar milieu and to minimise ongoing oxidative lung damage. Experimental novel therapeutic modalities manipulating cellular signalling pathways and apoptotic transcription factors (NOX inhibition, ERK 1/2 phosphorylation, JNK and p38 MAPK inhibition, caspase 3/9 inhibition, and NFκβ inhibition) with the augmentation of defence mechanisms through improving survival genes such as Nrf2 and AP-1 are important future research targets. Inflammasomes are cytosolic receptors and sensors that are involved in innate immune response, and their activation has been implicated in the pathogenesis of both hyperoxic lung injury and SARS-CoV-2 disease severity. Specifically targeting inflammasome activation may ameliorate lung injury and is subject to multiple clinical trial in COVID-19 patients [157,158].

## 6. Steroids for COVID-19 Patients Requiring Oxygen Therapy

There is established evidence that corticosteroids are beneficial for patients with COVID-19 lung disease requiring oxygen [159]. However, the main question remains: what is the pathophysiological basis for the clinical efficacy demonstrated by dexamethasone in COVID-19 patients requiring oxygen? The established paradigm suggests that dexamethasone acts as a potent anti-inflammatory agent to mitigate alveolar inflammation during the COVID-19 “cytokine storm”. However, neither the RECOVERY nor the REMAP-CAP studies performed any phenotypic evaluation to pre-characterise patients with classical features of the “cytokine storm” observed in some hospitalised COVID-19 patients [160,161,162]. Additionally, it is unclear how clinical signs of any cytokine storm correlate to the degree of alveolar damage and oxygen requirement. It is also uncertain if the progression into AHRF and the need for mechanical ventilation is directly attributable to the degree of inflammatory response or the “cytokine storm” seen in COVID-19 patients. Nevertheless, the cytokine response in COVID-19 patients is rather modest in comparison with other inflammatory conditions such as cytokine release syndrome (CRS) seen during chimeric antigen receptor (CAR) T-cell therapy, sepsis, and non-COVID-19-related ARDS [163]. Consequently, the beneficial effect of corticosteroids is perhaps mediated by other possible mechanisms that have not yet been explored. Moreover, the clinical efficacy is only demonstrated in patients requiring oxygen. Although the use of inhaled corticosteroids is associated with the downregulation of ACE2 receptors and may reduce the susceptibility to SARS-CoV-2 infection, the impact of dexamethasone on ACE-2/ANGII/ANG (1-7)/Mas levels and receptor mass in not known. Corticosteroids may have a direct effect on coagulation pathways including reductions in VWF and fibrinogen levels, which have not been evaluated in COVID-19 patients [164,165]. Alternative hypotheses for the beneficial effects of corticosteroids in patients requiring oxygen may include (1) a direct mitigating effect on oxidative stress, (2) the modulation of ACE2/ANGII/ANG (1-7)/Mas receptors, (3) an increase in surfactant synthesis or the stimulating differentiation of AT-II cells, and (4) an impact on the modulation of immunothrombosis. No clinical trials so far have evaluated the effect of steroids on specific haematological/prothrombotic variables on thrombotic tendencies.

## 7. Conclusions

The excessive use of oxygen to mitigate systemic hypoxia in patients with severe COVID-19 pneumonia may be associated with worsening lung injury and progression into the development of severe ARDS. Alveolar hyperoxia can lead to epithelial cell apoptosis, alterations in surfactant metabolism, increased oxidative stress from ROS, the downregulation of ACE2 receptors leading to a proinflammatory state from accumulation of ANGII, and lung and gut dysbiosis predisposing to secondary bacterial infections. Preventative strategies to minimise hyperoxic lung damage with the lowest possible FiO_2_, the use of CPAP, and maintaining adequate oxygen delivery by cardiovascular resuscitative measures may be of value from outset. Targeted therapies to moderate microangiopathy in combination with pulmonary vasodilators at this stage may be of value than a high FiO_2_, and these interventions are currently being evaluated in ongoing clinical trials. However, in situ microthrombosis may be problematic to lyse due to the inability of the anticoagulant to reach the vasoconstricted thrombosed site. iNO is often used as a rescue pulmonary vasodilator during AHRF and ARDS. A trial of pulmonary vasodilators as a preventative strategy may help to improve shunt fraction and to minimise oxygen-induced lung damage. Prolonged courses of prone positioning in responders may minimise the requirement for a high FiO_2_. There are several promising therapeutic targets in animal models of hyperoxic acute lung injury, including statins, surfactant therapy, antioxidant therapies, and omeprazole which warrants further evaluation.

## Figures and Tables

**Figure 1 medsci-11-00070-f001:**
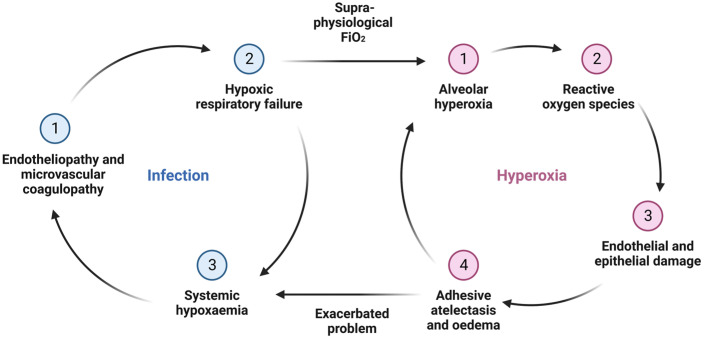
The SARS-CoV-2 infection exacerbated by alveolar hyperoxia leading to worsening systemic hypoxemia.

**Figure 2 medsci-11-00070-f002:**
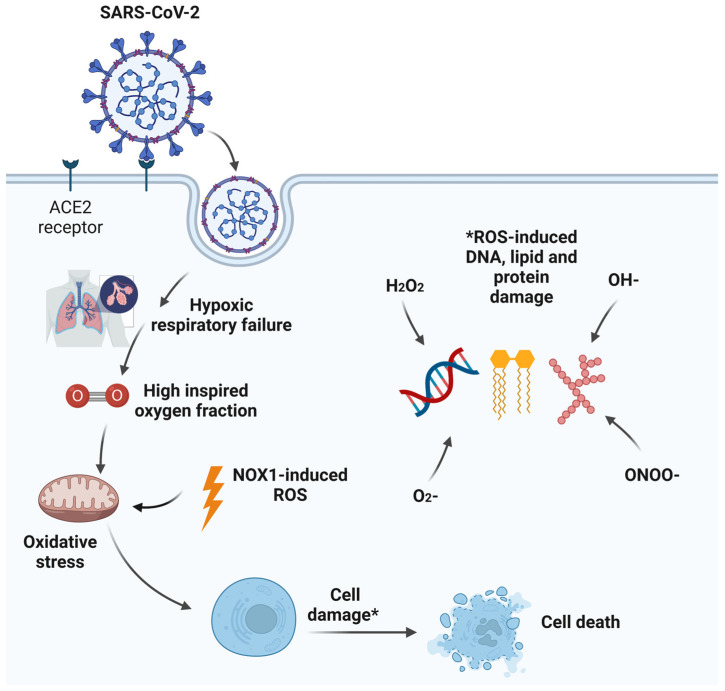
Increased oxidative stress as a result of a combination of SARS-CoV-2 viral infection and alveolar hyperoxia. * Reactive oxygen species.

**Figure 3 medsci-11-00070-f003:**
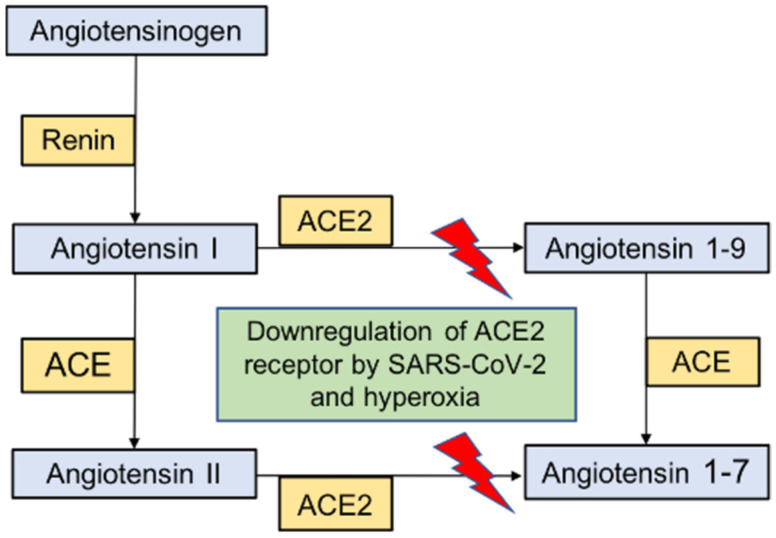
The renin–angiotensin/Mas system and the potential consequences of downregulation of ACE2 receptors.

**Table 1 medsci-11-00070-t001:** The opposing effects of angiotensin II and angiotensin (1-7).

Angiotensin II	Angiotensin 1-7
Vasoconstrictor	Vasodilator
Potent stimulator of NADPH oxidase	Suppress the NADPH oxidase activity
Increase reactive oxygen and nitrogen species	Reduce reactive oxygen and nitrogen species
Increase oxidative stress	Reduce oxidative stress
Proinflammation	Anti-inflammation
Profibrotic	Anti-fibrotic

## Data Availability

Not applicable.

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
