# Peer review of "Alveolar Hyperoxia and Exacerbation of Lung Injury in Critically Ill SARS-CoV-2 Pneumonia"

_medsci, 2023, doi:10.3390/medsci11040070_

Round 1
Reviewer 1 Report
Comments and Suggestions for Authors
Well costructed review, excellent figures.
Author Response
We are greateful for the reviewer's comments.
Reviewer 2 Report
Comments and Suggestions for Authors
Dushianthan et al. provide a comprehensive review of alveolar hyperoxia and the exacerbation of lung injury in SARS-CoV-2 pneumonia. They provide physiologic and pathophysiologic concepts and information obtained from SARS-CoV-2 patients to demonstrate the effects of hyperoxia-induced lung injury, particularly the key role played by reactive oxygen species augmenting the effects of viral infection. In addition, vascular/endothelial abnormalities leading to the formation of microthrombi are included. Strategies to minimize alveolar hyperoxia and particular measures to optimize oxygenation are discussed. The authors should be congratulated for putting together a tour de force review with over 160 references cited. The review is well-written, and the sections are appropriately designated. The following issues are missing or not well discussed in this manuscript and should be included.
1. The authors do not discuss age-dependent differences in responses to hyperoxia. For example, neonates are more tolerant of hyperoxia than adults (PMID: 16781448).
2. There is no mention of the activation of inflammasomes by SARS-CoV-2. There is extensive literature on this, with some trials suggesting improved outcomes of inflammasome inhibition in human clinical trials (PMID: 37703588, 37515138, 37360532).
3. Components of the extracellular matrix are key in the development of inflammation, and several ECM molecules have been implicated in COVID-19 (PMID: 36750167, 36687194, 35675639).
4. Nuclear Factor kappa B (NFkB) is defined incorrectly throughout the manuscript as nuclear factor kappa-beta.
Author Response
Dushianthan et al. provide a comprehensive review of alveolar hyperoxia and the exacerbation of lung injury in SARS-CoV-2 pneumonia. They provide physiologic and pathophysiologic concepts and information obtained from SARS-CoV-2 patients to demonstrate the effects of hyperoxia-induced lung injury, particularly the key role played by reactive oxygen species augmenting the effects of viral infection. In addition, vascular/endothelial abnormalities leading to the formation of microthrombi are included. Strategies to minimize alveolar hyperoxia and particular measures to optimize oxygenation are discussed. The authors should be congratulated for putting together a tour de force review with over 160 references cited. The review is well-written, and the sections are appropriately designated. The following issues are missing or not well discussed in this manuscript and should be included.
We are extremely greateful for reviewers comments. Please see the reponse, below.
- The authors do not discuss age-dependent differences in responses to hyperoxia. For example, neonates are more tolerant of hyperoxia than adults (PMID: 16781448). Response: we have now included a sentence to address this comment (page 4, 158-161).
- There is no mention of the activation of inflammasomes by SARS-CoV-2. There is extensive literature on this, with some trials suggesting improved outcomes of inflammasome inhibition in human clinical trials (PMID: 37703588, 37515138, 37360532). Response: We have now included a sentence to address this comment (page 13, 564-568).
- Components of the extracellular matrix are key in the development of inflammation, and several ECM molecules have been implicated in COVID-19 (PMID: 36750167, 36687194, 35675639). Response: We agree with the reviewer, that ECM remodelling has been attributed to COVID-19 pathogenesis. We have included the certain components of ECM that have been already established as contributing factors for COVID-19 related microangiopathy.
- Nuclear Factor kappa B (NFkB) is defined incorrectly throughout the manuscript as nuclear factor kappa-beta. Response: We have corrected these as suggested.
Reviewer 3 Report
Comments and Suggestions for Authors
Your review provides a fine and complete overview of the effects of dysoxia and dysoxemia, in general and in COVID-19. Presentation, though, could improve, as the review jumps (too) often from ‘non-COVID’ to ‘COVID’, at times even within the same paragraph, causing confusion, at least to me.
Therefore, in general, I would suggest to first discuss what we know from pre-COVID (animal experiments, clinical studies, direct effects of (high) oxygen (levels) on lung tissue, indirect effects (of high blood and tissue levels of) oxygen, etc.), then discuss why hyperoxia happened so often in COVID-19, and why COVID-19 is different, and then discuss (differences in) therapies (and whether or not the consequences are different between classic ARDS (or AHRF) patients and COVID-19 ARDS (or AHRF) patients. You partly already did so, but ‘skeptical’ readers may say ‘there is nothing new about COVID, it is ARDS as we know it, or at best an (etiologic) subphenotype’, and to be honest with you I partly agree — so make your point way better by not having information on differences between these two entities (with regard to oxygen) snowed under?
Many paragraphs are ‘long’ and not so much to the point. For instance … Introduction is quite long; first paragraph is nice, but not needed in this review; I suggest to remove and start Introduction with the second paragraph — for that same reason delete first sentences of the third paragraph on the UK AND Italian cohorts/report — and at the end, for reasons explained above, announce better what you are going to do, what you will discuss in what order. There are more examples of this throughout the whole manuscript.
Opposite, there are also parts that are short, if not too short. For instance …, in the part on ‘therapies’, text and reasoning becomes a bit ‘sloppy’ here and there with too many ‘easy’ statements — some things are only slightly touched upon (e.g., APRV) and there you scratch more the surface than really discussing how e.g., to set a ventilator in the context of hypoxemia (and classic ARDS versus COVID ARDS); I did miss a chapter on HFNO too (where the problem of ‘oxygen toxicity’ may even be more important than with ventilation — see e.g., Hyperoxia-induced lung injury in acute respiratory distress syndrome: what is its relative impact? Lilien TA, van Meenen DMP, Schultz MJ, Bos LDJ, Bem RA. Am J Physiol Lung Cell Mol Physiol. 2023 Jul 1;325(1):L9-L16. doi: 10.1152/ajplung.00443.2022. Epub 2023 May 2. PMID: 37129255) [I am biased, I served as a co-author of this paper—you do not need to cite it, but here we discuss the risks of hyperoxia too, it may help)
Also lastly, in therapies, you can try to delineate better what is different between ‘ARDS as we know it’ and COVID ARDS? Some therapies (corticosteroids) are also given to non-COVID ARDS patients: is it really different, and then: how is it different?
Let me assure you, I like your manuscript - but the message could improve by reorganizing it, and delving deeper, focussing more on differences between classic ARDS (AHRF) and COVID ARDS (AHRF), otherwise your paper will end up at the pile of ‘just another paper on COVID’?
Author Response
We are grateful for the reviewer's commnets and please see the reponse below.
1. Your review provides a fine and complete overview of the effects of dysoxia and dysoxemia, in general and in COVID-19. Presentation, though, could improve, as the review jumps (too) often from ‘non-COVID’ to ‘COVID’, at times even within the same paragraph, causing confusion, at least to me.
Response: Thank you. We have reviewed the whole manuscript to address any ambiguity. The aim of the review is to provide evidence wheere available. In the absence of any specific COVID-19 related literature, conclusions were drawn from ARDS and critical illnes. We have clarified this thoughout the manuscript.
2. Therefore, in general, I would suggest to first discuss what we know from pre-COVID (animal experiments, clinical studies, direct effects of (high) oxygen (levels) on lung tissue, indirect effects (of high blood and tissue levels of) oxygen, etc.), then discuss why hyperoxia happened so often in COVID-19, and why COVID-19 is different, and then discuss (differences in) therapies (and whether or not the consequences are different between classic ARDS (or AHRF) patients and COVID-19 ARDS (or AHRF) patients. You partly already did so, but ‘skeptical’ readers may say ‘there is nothing new about COVID, it is ARDS as we know it, or at best an (etiologic) subphenotype’, and to be honest with you I partly agree — so make your point way better by not having information on differences between these two entities (with regard to oxygen) snowed under? Many paragraphs are ‘long’ and not so much to the point. For instance … Introduction is quite long; first paragraph is nice, but not needed in this review; I suggest to remove and start Introduction with the second paragraph — for that same reason delete first sentences of the third paragraph on the UK AND Italian cohorts/report — and at the end, for reasons explained above, announce better what you are going to do, what you will discuss in what order. There are more examples of this throughout the whole manuscript.
Response: We feel that the introduction paragraph is important to provide a summary of COVID-19. We have now split the introduction section into two. The first section deals with COVID-19 and the following section concentrates on COVID-19, oxygen therapy and alveolar hyperoxia. The sentences relate to the incidence and degree of oxygen therapy in COVID-19 is crucial for this review. The manuscript is organised in such a way to deal with important sections: COVID-19, COVID-19 and oxygen therapy, consequence of alveolar and systemic hyperoxia, SARS-COV-2 endotheliopathy and how to minimise hyperoxia. The section of ubiquitous use of oxygen may be a problem- pre COVID- 19 studies, deals with animal studies first, followed by human studies and randomised trials as per reviewer’s suggestion.
3. Opposite, there are also parts that are short, if not too short. For instance …, in the part on ‘therapies’, text and reasoning becomes a bit ‘sloppy’ here and there with too many ‘easy’ statements — some things are only slightly touched upon (e.g., APRV) and there you scratch more the surface than really discussing how e.g., to set a ventilator in the context of hypoxemia (and classic ARDS versus COVID ARDS); I did miss a chapter on HFNO too (where the problem of ‘oxygen toxicity’ may even be more important than with ventilation — see e.g., Hyperoxia-induced lung injury in acute respiratory distress syndrome: what is its relative impact?Lilien TA, van Meenen DMP, Schultz MJ, Bos LDJ, Bem RA. Am J Physiol Lung Cell Mol Physiol. 2023 Jul 1;325(1):L9-L16. doi: 10.1152/ajplung.00443.2022. Epub 2023 May 2. PMID: 37129255) [I am biased, I served as a co-author of this paper—you do not need to cite it, but here we discuss the risks of hyperoxia too, it may help)
Response: The aim of this review is to explore potential mechanisms of hyperoxia induced lung injury and provide a succent summary of measures how to minimise it in COVID-19, drawing similarities from pre-existing studies of critical illness. While it would be good to provide detailed sections on the similarities and differences between ARDS and CARDS and therapeutic options, this was not the aim of this manuscript. We did not specifically address HFNO and NIV/CPAP as these will fall into the generic section of permissive hypoxemia.
4. Also lastly, in therapies, you can try to delineate better what is different between ‘ARDS as we know it’ and COVID ARDS? Some therapies (corticosteroids) are also given to non-COVID ARDS patients: is it really different, and then: how is it different?
Response: Thank you for the reviewer’s comments. The manuscript was not aimed to provide detailed description and the comparisons between of ARDS and CARDS.
5. Let me assure you, I like your manuscript - but the message could improve by reorganizing it, and delving deeper, focussing more on differences between classic ARDS (AHRF) and COVID ARDS (AHRF), otherwise your paper will end up at the pile of ‘just another paper on COVID’?
Response: Thank you. We have attempted to provide a detailed pathophysiological process that may underpin hyperoxia-induced lung injury in critically ill COVID-19 patients. While it is tempting to explore a more detailed, in-depth analysis of pathophysiological features and treatment strategies differentiating ARDS and CARDS, this was not the aim of this review and is beyond the scope of this intended manuscript.
Round 2
Reviewer 2 Report
Comments and Suggestions for Authors
The authors have adequately revised the manuscript.
Reviewer 3 Report
Comments and Suggestions for Authors
I have the impression you did not try to shorten the manuscript, i.e., to be more concise - it is a choice, of course, but lengthy papers are not read well, and message get snowed under
as one example (just one but there are many), section 1 is not needed, you can simply delete it, really.
structure can still be improved a lot